# Homogeneous Zero-Index Thermal Metadevice for Thermal Camouflaging and Super-Expanding

**DOI:** 10.3390/ma16103657

**Published:** 2023-05-11

**Authors:** Huagen Li, Kaipeng Liu, Tianfeng Liu, Run Hu

**Affiliations:** 1Department of Electrical and Computer Engineering, National University of Singapore, Singapore 117583, Singapore; 2State Key Laboratory of Coal Combustion, School of Energy and Power Engineering, Huazhong University of Science and Technology, Wuhan 430074, China

**Keywords:** heat transfer, thermal metamaterials, thermal camouflage

## Abstract

The infinite effective thermal conductivity (IETC) can be considered to be an equivalence of the effective zero index in photonics. A recent highly rotating metadevice has been discovered to approach near IETC, subsequently demonstrating a cloaking effect. However, this near IETC, related to a rotating radius, is quite inhomogeneous, and the high-speed rotating motor also needs a high energy input, limiting its further applications. Herein, we propose and realize an evolution of this homogeneous zero-index thermal metadevice for robust camouflaging and super-expanding through out-of-plane modulations rather than high-speed rotation. Both the theoretical simulations and experiments verify a homogeneous IETC and the corresponding thermal functionalities beyond cloaking. The recipe for our homogeneous zero-index thermal metadevice involves an external thermostat, which can be easily adjusted for various thermal applications. Our study may provide meaningful insights into the design of powerful thermal metadevices with IETCs in a more flexible way.

## 1. Introduction

Thermal metamaterials have achieved a plethora of unusual temperature profiles based on the transformation optics theory [1,2,3,4] and scattering cancelling technology [5,6], and various thermal metadevices, such as the switchable [7], inverse thermal [8], doublet [9], tunable [10,11], adaptive rotating [12], and printable freeform schemes [13], have thereby been proposed and achieved, resulting in various thermal management functionalities. Among them, the effective thermal conductivity (ETC) serves as one of the key parameters [14]. However, the capability of thermal metamaterials to realize various thermal functions is very narrow due to the limited range of thermal conductivities [15]. Although many human-made materials have acquired high thermal conductivities, such as graphene at 5300 W m^−1^ K^−1^ [16], they are still quite low in contrast to the requirement of an infinite thermal conductivity in thermal metamaterials. Notably, the performance of the traditional thermal expander [17,18], thermal illusion [19], and camouflage [20,21,22,23] also has been limited by the range of ETCs. Meanwhile, compared with commonly used materials, such as aluminum and copper, their fabrication processes are usually quite complex [14].

Recently, to solve the above problems, a bilayer thermal cloak introducing a convective element has been reported to have the capability to acquire an inhomogeneous near IETC [15]. Specifically, for this bilayer thermal cloak, there are two concentric shells with two different radii. It can avoid the distortion of the external field with a cloaking effect when the corresponding matching function is satisfied. Moreover, the infinite thermal conductivity in heat conduction happens to be considered to be an equivalence of the zero index in photonics [24,25,26,27,28,29,30,31]. However, the high-speed rotating near-zero-index scheme needs a high energy input, and the obtained near IETC is quite inhomogeneous, resulting in the limitation of the thermal application scenarios [32]. In addition, solving the inhomogeneous problem in both the optical and thermal fields is usually a challenging task [10]. Thus, it is quite nontrivial and challenging to solve the high-energy-consumption and inhomogeneous problems in a zero-index scheme. On the other hand, this rotating scheme currently depends on an actual infinite fluid velocity to obtain an IETC, which is impossible to exactly obtain in practice [15]. Additionally, the evolution of homogeneous zero-index thermal metadevices for camouflaging and super-expanding has not been experimentally demonstrated. Thereby, the realization of homogeneous zero-index thermal camouflaging and super-expanding applications via a more feasible and flexible method is worth being further investigated.

Herein, we propose and realize a homogeneous zero-index thermal camouflage metadevice and a zero-index super thermal expander via a simple external thermostat, such as thermostatic sinks, rather than high-speed rotation both theoretically and experimentally. Meanwhile, the height change of the ultra-thin hollow wall in our scheme can also be utilized to obtain a tunable ETC (TETC). Moreover, it is quite easy to tune these out-of-plane conditions by just adjusting the thermostatic sink or height of the ultra-thin hollow wall, resulting in a feasible and flexible pathway to realize various thermal functions beyond cloaking. Our work is expected to provide meaningful insights into the design of robust thermal metadevices, which may open more strategies for manipulating the heat flux in a more flexible way.

## 2. Theoretical Analysis

In photonics, when the value of either permittivity *ε* or permeability *μ* is zero (known as zero-index media); the Helmholtz equation, $\left(\nabla^{2}+\varepsilon \mu \omega^{2}\right)E=0$, degenerates to a Laplacian equation, $\nabla^{2}E=0$, where *ω* and **E** are the angular frequency and the electric field, respectively. Such zero-index metamaterials have induced considerable interesting phenomena, and, simultaneously, the inhomogeneous issue in optical zero-index media usually exists, resulting in the further applications of these zero-index metamaterials [24,25,26,27,28,29,30,31]. Similarly, in heat diffusion, the Fourier-transformed heat conduction equation, $\left(\nabla^{2}-i\omega \rho c_{p}/\kappa \right)T=0$, also degenerates to a Laplacian equation when the thermal conductivity κ approaches IETC, where *ρ*, *c_p_*, and *T* are the density, heat capacity, and temperature, respectively. In diffusion, an IETC, related to the optical zero-index concept, can be realized through different strategies as shown in Figure 1. Herein, the traditional inhomogeneous near-zero-index scheme was realized by highly rotating the on-demand region (Figure 1a) [15]. Similarly, the out-of-plane modulation scheme (Figure 1b), in principle, can introduce a homogeneous IETC, indicating a more flexible way to obtain a homogeneous IETC. Thus, the homogeneous IETC can be introduced in practice and can be rendered an equivalence of the zero index in photonics. Additionally, we could also realize a continuous tunable ETC (TETC) via the height change of the ultra-thin hollow wall in the central part. Finally, a homogeneous IETC and TETC could be achieved by combining a thermostatic sink and the ultra-thin hollow wall together for diverse thermal applications (Figure 1c), and, furthermore, the real photo of the proposed metadevice is shown in Figure 1d.

Considering a 2D ultra-thin elliptical hollow structure embedded in the center of a rectangle background as shown in Figure 2a, the outer semiaxes of the 2D ultra-thin elliptical hollow structure were expressed as $r_{ei}$ (*i* = 1, 2). According to the related references [33,34,35,36], we could obtain the heat conduction equation in the elliptic coordinate (ξ,θ) to be
(1)$$\nabla^{2}T=\frac{1}{\rho^{2}}\left(\frac{\partial^{2}T}{\partial \xi^{2}}+\frac{\partial^{2}T}{\partial \theta^{2}}\right)=0$$
where ρ and the relationship between the Cartesian coordinates and the elliptic coordinates are $\rho =\sqrt{r_{e1}^{2}-r_{e2}^{2}}\sqrt{\sinh^{2}\xi +\cos^{2}\theta }$ and $\{\begin{array}{l} x=\sqrt{r_{e1}^{2}-r_{e2}^{2}}\cosh\xi \cos\theta \\ y=\sqrt{r_{e1}^{2}-r_{e2}^{2}}\sinh\xi \sin\theta \end{array}$, respectively. Then, the corresponding boundary conditions in this 2D ultra-thin elliptical hollow structure were
(2)$$\{\begin{array}{l} T_{e}\;\mathrm{is}\;\mathrm{finite}\;\\ T_{b}(\rho \to \infty )={\vec{H}}_{i}\cdot {\vec{x}}^{i}\\ T_{e}(\xi )=T_{u}(\xi )\\ -{\kappa_{e}\frac{\partial T_{e}}{\partial \xi }|}_{\xi =\frac{1}{2}\ln\left(\frac{r_{e1}+r_{e2}}{r_{e1}-r_{e2}}\right)}=-{\kappa_{b}\frac{\partial T_{b}}{\partial \xi }|}_{\xi =\frac{1}{2}\ln\left(\frac{r_{e1}+r_{e2}}{r_{e1}-r_{e2}}\right)} \end{array}$$
where $\kappa_{b}$ and $\kappa_{e}$ are the thermal conductivity of this 2D ultra-thin elliptical hollow part and the background, respectively. The corresponding temperature fields of the ultra-thin hollow structure and background are $T_{e}$ and $T_{b}$. Meanwhile, ${\vec{H}}_{i}$ is denoted as the scalar value of the thermal gradient field along the two different axes in the Cartesian coordinates. Then, the general solution of temperature $T_{e}$ in the 2D ultra-thin elliptical hollow structure could be denoted as:(3)$$T_{e}=C-\frac{\kappa_{b}}{S_{ei}\kappa_{e}+\left(1-S_{ei}\right)\kappa_{b}}{\vec{H}}_{i}\cdot {\vec{x}}^{i}$$
where ${\vec{x}}^{i}$ (*i* = 1, 2) denote the unit vectors in the Cartesian coordinates, *C* is the reference constant temperature, and *S_ei_* (*i* = 1, 2) are the shape parameters of the 2D ultra-thin hollow structure along the two different axes in the Cartesian coordinates, i.e.,
(4)$$S_{ei}=\frac{r_{e1}r_{e2}}{2}\times \int_{0}^{\infty }\frac{dl}{\left(l+r_{ei}^{2}\right)\sqrt{\left(l+r_{e1}^{2}\right)\left(l+r_{e2}^{2}\right)}}$$

Thus, when the actual thermal conductivity $\kappa_{e}$ of the 2D ultra-thin elliptical hollow structure was infinite, it was easy to see that *T_e_* always approached a constant temperature *C* as shown in Figure 2b. However, the actual thermal conductivities of naturally occurring materials are usually smaller than 400 W m^−1^ K^−1^ [10]. For physicists, if the ETC $\kappa_{e}^{eff}$ is infinite, the heat flux will not diverge, and it can realize a constant temperature distribution. In other words, a homogeneous IETC can be obtained when any object is in contact with a constant temperature boundary condition as shown in Figure 2c.

For practical implementation, we extended this 2D ultra-thin elliptical hollow structure to a 3D version, and the height of the ultra-thin hollow wall was found to have a significant influence on its in-plane ETC, yielding a tunable ETC (TETC). The 2D and 3D schematics of the circular ultra-thin hollow wall embedded in the background are shown in Figure 2d,e, respectively. Then, the TETC could be theoretically derived as (see Appendix A):(5)$$k_{u}^{eff}=\frac{-k_{u}m_{2}\tanh\left[h/\left(r_{u}-\sigma \right)\right]+3k_{b}m_{1}\left(r_{u}-\sigma \right)}{3m_{1}d\left(r_{u}-\sigma \right)}$$
where σ and d are the thickness of the ultra-thin hollow wall and background, respectively. $r_{u}$ and *h* are the radius and height of the ultra-thin hollow wall. $\kappa_{u}$ and $\kappa_{b}$ are the corresponding thermal conductivities of the ultra-thin elliptical hollow wall and background. $m_{1}$ and $m_{2}$ are related to $r_{u}$ and σ (see Appendix A). With a 3D ultra-thin hollow wall, we could achieve the homogeneous IETC experimentally by inserting the bottom of the hollow wall into a thermostatic sink, and we also obtained a TETC by adjusting the depth of the submerged hollow wall as shown in Figure 2f.

Furthermore, to demonstrate the potential applications of our homogeneous zero-index thermal metamaterials, we realized a homogeneous zero-index thermal camouflage device and super thermal expander as the corresponding evolutions of the traditional zero-index thermal metadevice. Specifically, the expected reference object and traditional thermal expander [17] are shown in Figure 2g,i, respectively. Eventually, the homogeneous zero-index thermal camouflage and super thermal expander effect could be tactfully formed by combining the on-demand structure around the ultra-thin wall as shown in Figure 2h,j, leading to the restoration of the failed thermal functionalities under a super highly conductive background.

## 3. Results and Discussions

To verify the advantages of above two homogeneous zero-index thermal metadevices, the corresponding simulations were carried out with the software COMSOL MULTIPHYSICS 5.6 as shown in Figure 3. Herein, the thermal conductivity $\kappa_{b}$ of the background was set up to be 400 W m^−1^ K^−1^, and the geometric parameters *a* and *L* were 30 mm and 40 mm, respectively. The circular case is demonstrated in Figure 3a,b. Obviously, it is easy to observe that there was an exact equivalence between the identical thermal signature in the external field as shown in Figure 3a,b, indicating the effective thermal camouflage effect. Therefore, the camouflage effect was excellent, and, furthermore, according to their almost identical temperature curves, the effect was further verified. Notably, we just applied a thermostatic sink in the central part with an ultra-thin hollow wall structure to realize a homogeneous IETC, and then it only required the outer layer with the thermal conductivity $\kappa_{v}$ = 196 W m^−1^ K^−1^_,_ which is much lower than 400 W m^−1^ K^−1^. Additionally, this scheme could also be used to obtain a homogeneous zero-index super thermal expander as shown in Figure 3d. In contrast to the traditional thermal expander, as shown in Figure 3c, this homogeneous zero-index super thermal expander exhibited more straight isotherms, indicating excellent thermal expanding performance. In addition, the related simulated temperature profiles of the two different homogeneous zero-index thermal metadevices are depicted in Figure 3e and 3f, respectively. Notably, the zero-index thermal camouflage effect was quite excellent based on the almost identical temperature curves of the equivalent expected reference and zero-index thermal camouflage. Moreover, from the results shown in Figure 3f, it is easy to observe that the expander effect was also excellent, and, furthermore, according to the evolutions of the temperature curves, a good effect was further verified. In brief, the zero-index super thermal expander can quickly provide a homogeneous temperature gradient, which is much better than the traditional thermal expander.

To further confirm the performance of the homogeneous zero-index thermal camouflage and super thermal expander, the corresponding experimental demonstration was carried out. Herein, the integrated fabrication technology was used to ensure these thermal metadevices were without weld joints. Meanwhile, the designed thermal conductivity of the outer layer was obtained through air holes drilled into the copper plate. Then, an external thermostatic sink with a medium temperature was placed to contact the ultra-thin hollow wall so as to obtain a constant temperature boundary condition, resulting in a homogeneous IETC with this simpler design. Different from other schemes that need a complicated design, this scheme provides a more feasible pathway, which may be beneficial for various practical applications. Additionally, these four samples were covered with transparent plastic to decrease the influence of the environment.

Furthermore, a FLIR E60 infrared camera was used to observe the temperature fields of these thermal metadevices as shown in Figure 4. Obviously, the experimental results, as shown in Figure 4a–d, were almost the same as the corresponding numerical simulations above, further validating the capability of the homogeneous zero-index scheme to realize various evolutions of thermal metadevices. Likewise, the related measured temperature profiles of the two different homogeneous zero-index thermal metadevices are depicted in Figure 4e,f, respectively. In all, compared with the rotating scheme (see Appendix A), our homogeneous zero-index scheme provides advantages in terms of flexible thermal functionalities, low energy consumption, arbitrary structure availability, and wide application scenarios.

## 4. Conclusions

In summary, the homogeneous zero-index thermal camouflage metadevice and the zero-index super thermal expander, as two classes of metamaterials, have been proposed and realized, leading to a vital enhancement in molding heat fluxes at will. We have first introduced the coupled effects of the boundary conditions and heights via out-of-plane modulations in a diffusive system to macroscopically modulate the effective thermal conductivity. Moreover, we only needed an external thermostatic sink to realize the zero-index thermal camouflage or super thermal expander effect. Furthermore, these effects have been experimentally confirmed, demonstrating outstanding thermodynamic performance. Additionally, we have found out that the height change of our ultra-thin hollow wall in our scheme can also highly tune the ETC. This homogeneous zero-index paradigm with a simpler structure and more flexible mechanism provides more possibilities for the practical applications of various powerful thermal metadevices beyond cloaking. In all, this work expands the application scenarios of the zero-index concept and provides meaningful insights into the design of various zero-index thermal metadevices, which opens up more strategies for manipulating heat fluxes in a more flexible way.

## Figures and Tables

**Figure 1 materials-16-03657-f001:**
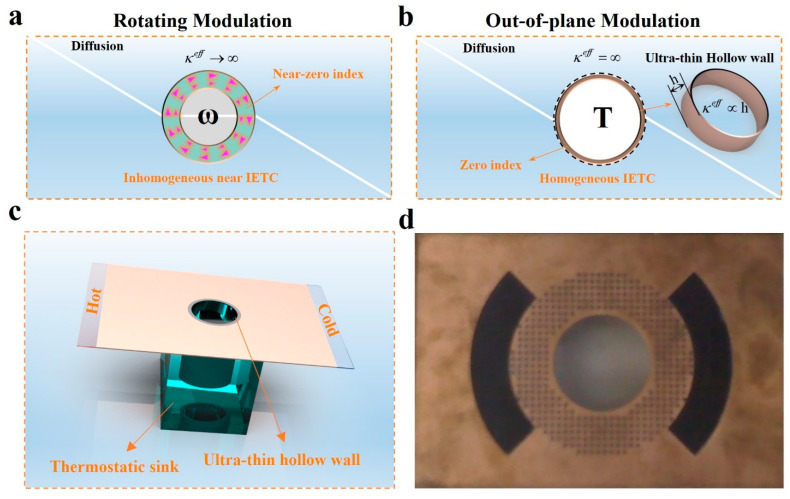
Origin of IETC, related to zero index, and correspondence between two distinct processes. (**a**) inhomogeneous near IETC realized through a rotating modulation. (**b**) Homogeneous IETC realized through the temperature change of ultra-thin hollow wall and TETC realized via the height change of ultra-thin hollow wall (out-of-plane modulations). (**c**) The schematic diagram of our homogeneous zero-index scheme via out-of-plane modulations. (**d**) The real photo (real scale) of the proposed metadevice.

**Figure 2 materials-16-03657-f002:**
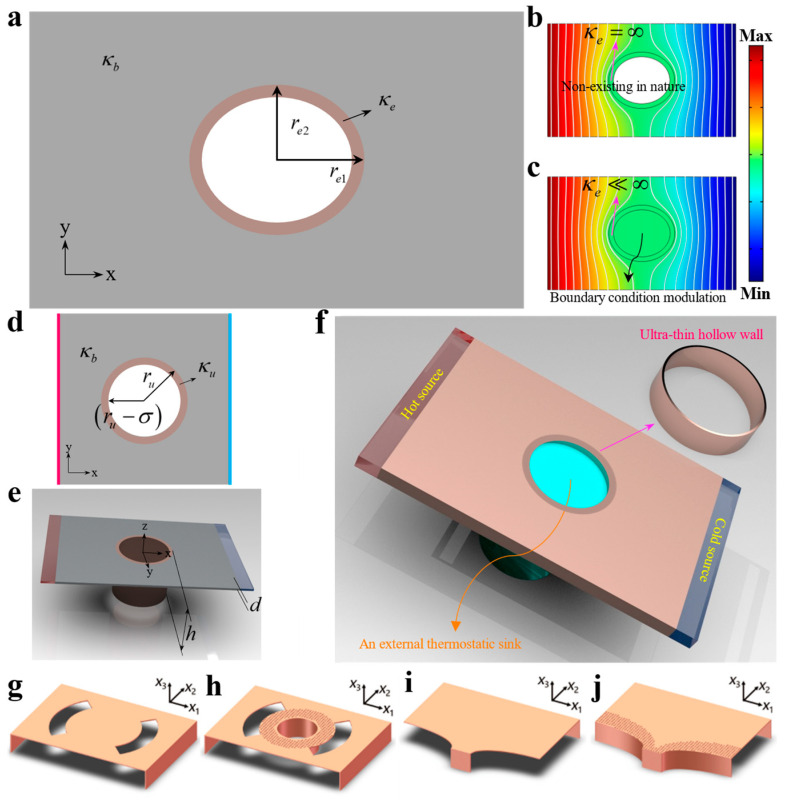
Schematic diagram and realization of homogeneous zero-index thermal camouflage and super thermal expander. (**a**) 2D Schematic of the elliptical ultra-thin hollow wall structure embedded in the background. (**b**) Temperature profile with the ultra-thin hollow structure ($\kappa_{e}$ = ∞) embedded in the background ($\kappa_{b}$ = 400 W m^−1^ K^−1^). (**c**) Temperature profile with the ultra-thin hollow structure($\kappa_{e}$ < ∞, actually set as 0.02 W m^−1^ K^−1^) and a constant temperature boundary condition (with temperature (Max + Min)/2) placed in the background. Here, Max and Min denote the temperatures of the left and right boundaries, respectively, and white lines represent isotherms. Schematic of 2D (**d**) and 3D (**e**) models of the circular ultra-thin hollow wall structure embedded in the background. (**f**) Sketch map of the homogeneous IETC and TETC realized by combining a thermostatic sink and the ultra-thin hollow wall. (**g**) Schematic diagram of the expected reference object. (**h**) Schematic diagram of homogeneous zero-index thermal camouflage. (**i**) Schematic diagram of traditional thermal expander. (**j**) Schematic diagram of homogeneous zero-index super thermal expander.

**Figure 3 materials-16-03657-f003:**
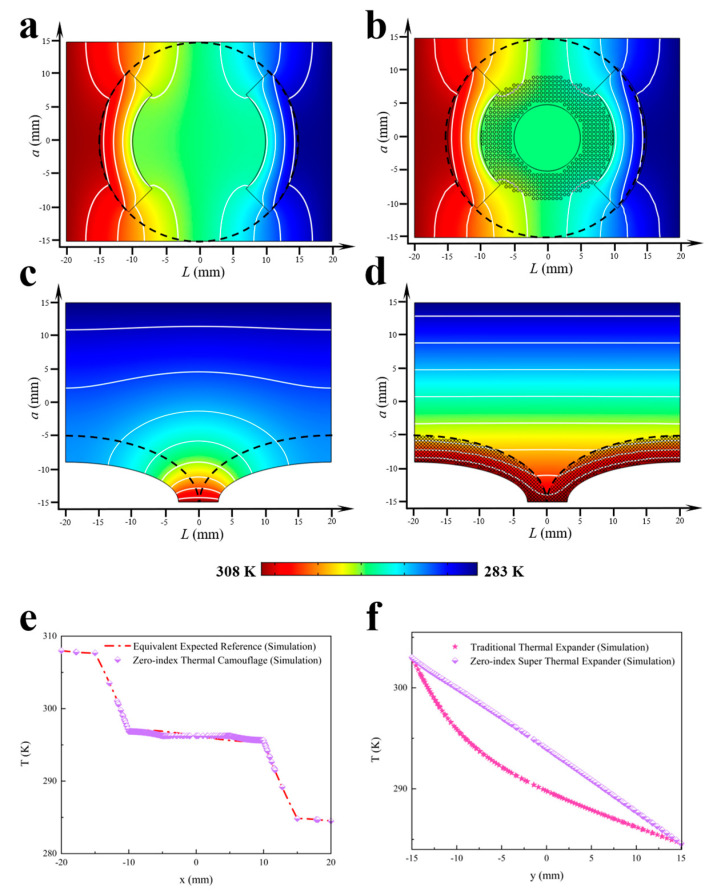
Simulations of zero-index thermal camouflage and super thermal expander. (**a**) Equivalent object corresponding to Figure 2g. (**b**) Homogeneous zero-index thermal camouflage device corresponding to Figure 2h. (**c**) Traditional thermal expander corresponding to Figure 2i. (**d**) Homogeneous zero-index super thermal expander device corresponding to Figure 2j. Isothermal lines are represented in white color on the panel. The highest and lowest temperatures were 308 K and 283 K, respectively. (**e**) The simulated temperature profiles of the equivalent expected objects and homogeneous zero-index thermal camouflage at the line y = 0 mm. (**f**) The simulated temperature profiles of the traditional thermal expander and homogeneous zero-index super thermal expander at the line x = 0 mm.

**Figure 4 materials-16-03657-f004:**
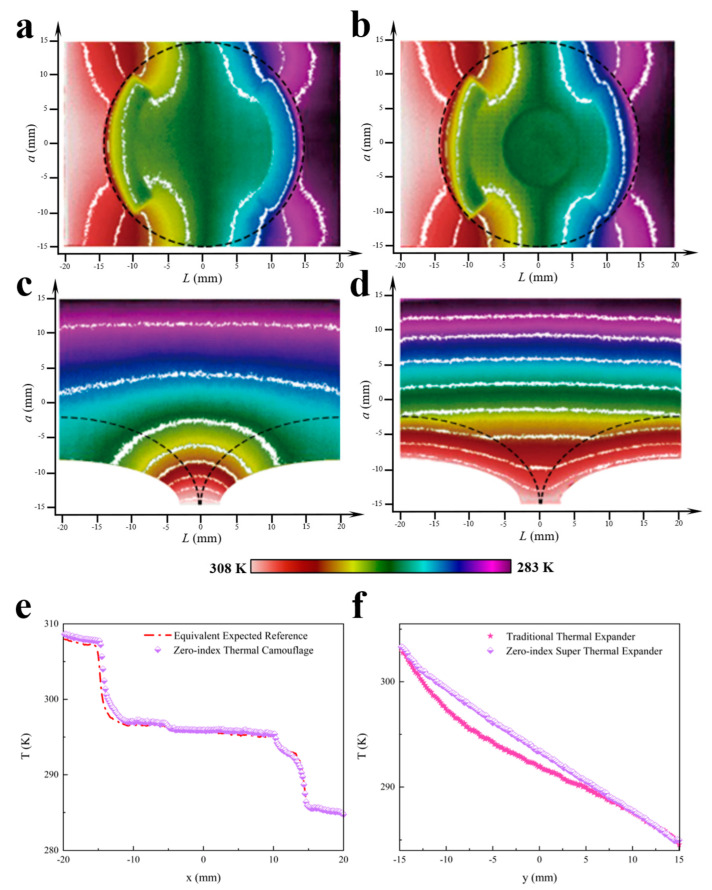
Experimental measurements of temperature profiles of various thermal metadevices. (**a**) Results for the equivalent expected objects in Figure 2g. (**b**) Results for the homogeneous zero-index thermal camouflage device in Figure 2h. (**c**) Results for the traditional thermal expander in Figure 2i. (**d**) Results for the homogeneous zero-index super thermal expander device in Figure 2j. Isothermal lines are represented in white color on the panel. The highest and lowest temperatures were 308 K and 283 K, respectively. (**e**) The temperature profiles of the equivalent expected objects and homogeneous zero-index thermal camouflage at the line y = 0 mm. (**f**) The temperature profiles of the traditional thermal expander and homogeneous zero-index super thermal expander at the line x = 0 mm.

## Data Availability

Not applicable.

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
