# Peer review of "Homogeneous Zero-Index Thermal Metadevice for Thermal Camouflaging and Super-Expanding"

_materials, 2023, doi:10.3390/ma16103657_

Round 1

Reviewer 1 Report

Comments to the Authors:

The present study proposes the design of a homogeneous zero-index thermal metadevice to realize infinite effective thermal conductivity aiming potential applications of robust thermal camouflage and super thermal expander. The introduced metadevice can be highly tuned, as analytically supported by both theoretical and experimental evaluation. Generally, the manuscript is informative, well-presented and the supplementary material assists the main findings. The content is suitable within the interest of Materials journal and is worth publishing after the authors consider the following suggestions:

Comment #1: Within the 1. Introduction section briefly include some sentences explaining the “cloaking effect” principles etc.

Comment #2: Within the 3. Results and discussions section “Obviously, the experimental results as shown in Fig. 4a-d are almost the same with the above corresponding numerical simulations. Likewise, the related measured temperature profiles for the two different homogeneous zero-index thermal metadevices are depicted in Fig. 4e and 4f, respectively. Obviously, it is easy to observe that the experimental results are almost the same with the corresponding simulation results, further validating the capability of the homogeneous zero-index scheme to realize various evolutions of thermal metadevice.”. Please merge and revise properly in order to avoid needless repetitions.

Comment #3: Within the Conclusion section “the homogeneous zero-index thermal camouflage metadevice and zero-index super thermal expander have been proposed and realized, as two classes of metamaterials with vital importance in molding heat flux at will”. Please purposely rephrase in order to meet the existing linguistic standards.

Comment #4: Please include within the main manuscript a real photo (real scale) of the proposed metadevice in Figure 1 or/and Figure 4.

Comment #5: It is strongly recommended to enrich the Conclusions section. The novelty and the future outlook of this work should be clearly highlighted.

Author Response

The present study proposes the design of a homogeneous zero-index thermal metadevice to realize infinite effective thermal conductivity aiming potential applications of robust thermal camouflage and super thermal expander. The introduced metadevice can be highly tuned, as analytically supported by both theoretical and experimental evaluation. Generally, the manuscript is informative, well-presented and the supplementary material assists the main findings. The content is suitable within the interest of Materials journal and is worth publishing after the authors consider the following suggestions:

Our reply: We thank the reviewer for the accurate summarization and comments on our work. These comments and suggestions are all helpful and valuable for improving our manuscript. We have carefully considered the comments with the detailed responses listed below.

Comment #1: Within the 1. Introduction section briefly include some sentences explaining the “cloaking effect” principles etc.

Our reply: We thank the referee for carefully raising this issue and giving valuable suggestion. We have included some sentences explaining the “cloaking effect” principles in the revised manuscript (page 1) and the details are shown listed below:

“Specially, for this bilayer thermal cloak, there are two concentric shells with two different radii. It can avoid the distortion of external field with a cloak effect when the corresponding matching function is satisfied.”

Comment #2: Within the 3. Results and discussions section “Obviously, the experimental results as shown in Fig. 4a-d are almost the same with the above corresponding numerical simulations. Likewise, the related measured temperature profiles for the two different homogeneous zero-index thermal metadevices are depicted in Fig. 4e and 4f, respectively. Obviously, it is easy to observe that the experimental results are almost the same with the corresponding simulation results, further validating the capability of the homogeneous zero-index scheme to realize various evolutions of thermal metadevice.”. Please merge and revise properly in order to avoid needless repetitions.

Our reply: We thank the referee for raising this concern and giving the useful suggestion. We have revised the needless repetition in the revised manuscript (page 4) and the details are shown listed below:

“Obviously, the experimental results as shown in Fig. 4a-d are almost the same with the above corresponding numerical simulations, further validating the capability of the homogeneous zero-index scheme to realize various evolutions of thermal metadevice. Likewise, the related measured temperature profiles for the two different homogeneous zero-index thermal metadevices are depicted in Fig. 4e and 4f, respectively.”

Comment #3: Within the Conclusion section “the homogeneous zero-index thermal camouflage metadevice and zero-index super thermal expander have been proposed and realized, as two classes of metamaterials with vital importance in molding heat flux at will”. Please purposely rephrase in order to meet the existing linguistic standards.

Our reply: We thank the referee for raising this concern and giving the useful suggestions. We have rephrased the related expressions in the revised manuscript (page 5) and the details are shown listed below:

“the homogeneous zero-index thermal camouflage metadevice and zero-index super thermal expander, as two classes of metamaterials, have been proposed and realized, leading to vital enhancement in molding heat flux at will.”

Comment #4: Please include within the main manuscript a real photo (real scale) of the proposed metadevice in Figure 1 or/and Figure 4.

Our reply: Thank you for providing the constructive suggestion. We have rephrased the real photo (real scale) of the proposed metadevice in Figure 1.

Comment #5: It is strongly recommended to enrich the Conclusions section. The novelty and the future outlook of this work should be clearly highlighted.

Our reply: Thank you for raising this good question. We have highlighted the novelty and outlook of this study in the revised manuscript (page 5) and the details are shown listed below:

“In summary, the homogeneous zero-index thermal camouflage metadevice and zero-index super thermal expander, as two classes of metamaterials, have been pro-posed and realized, leading to vital enhancement in molding heat flux at will. We have first introduced the coupled effects of the boundary conditions and heights via out-of-plane modulations into a diffusive system to macroscopically modulate the effective thermal conductivity. Moreover, we only need an external thermostatic sink to realize the zero-index thermal camouflage or super thermal expander effect. Furthermore, these effects are experimentally confirmed, demonstrating the outstanding thermody-namic performance. Besides, we find out that the height change of our ultra-thin hol-low wall in our scheme can also highly tune the ETC. This homogeneous zero-index paradigm with a simpler structure and more flexible mechanism provides more possi-bility for the practical applications of various powerful thermal metadevices beyond cloaking. In all, this work expands the application scenarios of the zeo-index concept and provides meaningful insights into the design of various zero-index thermal metadevices, which opens more strategies for manipulating the heat flux in a more flexible way.”

Reviewer 2 Report

+ What is nanomaterial used in this study? (it belongs to a special issue about nanomaterials)
+ Fig. 3a, b and Fig. 4a, b have to provide an extensive view of the l camouflage device similar to that of Fig. 2 b,c. I do not see any temperature profile in the background region which is important information to show whether the cloak design can work efficiently or not. 
+ The author should provide a comparison temperature profile of a cross-sectional line in the background. The computational domain and experimental device should be enlarged.

Round 2

Reviewer 2 Report

I cannot recommend it for publication. The main reason for my decision is that the author's result is incorrect. The curvature of the temperature profile in the background reveals that the thermal cloak cannot protect well the object in the cloaked region. The author has not addressed this issue adequately in the revision nor provided any convincing evidence or explanation to support their claim. Moreover, there is no significant change in the revised version in both experimental and simulation results compared to the original manuscript, except for some minor cosmetic edits.

Author Response

I cannot recommend it for publication. The main reason for my decision is that the author's result is incorrect. The curvature of the temperature profile in the background reveals that the thermal cloak cannot protect well the object in the cloaked region. The author has not addressed this issue adequately in the revision nor provided any convincing evidence or explanation to support their claim.

Our Reply: We thank the reviewer for the accurate summarization and comments on our work. These comments and suggestions are all helpful and valuable for improving our manuscript. We have carefully considered the comments with the detailed responses listed below.

Firstly, we thank the referee for carefully raising this issue and giving suggestions. We apologize for the inaccuracy of the cloak design as shown in Fig. R1. We are so sorry that we have not addressed this issue adequately in the last revision.

Fig. R1 The simulation result of the cloak design for the further application in a camouflage device.

Moreover, there is no significant change in the revised version in both experimental and simulation results compared to the original manuscript, except for some minor cosmetic edits.

Our Reply: We thank the referee for carefully pointing out this issue and giving suggestions. We apologize for the lack of significant change in the revised version. We have added some extra discussions in both experimental and simulation results.
